# FA Sliding as the Mechanism for the ANT1-Mediated Fatty Acid Anion Transport in Lipid Bilayers

**DOI:** 10.3390/ijms241813701

**Published:** 2023-09-05

**Authors:** Jürgen Kreiter, Sanja Škulj, Zlatko Brkljača, Sarah Bardakji, Mario Vazdar, Elena E. Pohl

**Affiliations:** 1Institute of Physiology, Pathophysiology, and Biophysics, Department of Biomedical Sciences, University of Veterinary Medicine, 1210 Vienna, Austria; jkreiter@stanford.edu (J.K.); sanja.skulj@vetmeduni.ac.at (S.Š.); sarah.bardakji@vetmeduni.ac.at (S.B.); 2Division of Organic Chemistry and Biochemistry, Rudjer Bošković Institute, 10000 Zagreb, Croatia; zlatko.brkljaca@selvita.com; 3Department of Mathematics, Informatics, and Cybernetics, University of Chemistry and Technology, 166 28 Prague, Czech Republic

**Keywords:** fatty acid cycling hypothesis, ADP/ATP carrier, AAC, uncoupling proteins, proton transport, fatty acids anion transport, arachidonic acid

## Abstract

Mitochondrial adenine nucleotide translocase (ANT) exchanges ADP for ATP to maintain energy production in the cell. Its protonophoric function in the presence of long-chain fatty acids (FA) is also recognized. Our previous results imply that proton/FA transport can be best described with the FA cycling model, in which protonated FA transports the proton to the mitochondrial matrix. The mechanism by which ANT1 transports FA anions back to the intermembrane space remains unclear. Using a combined approach involving measurements of the current through the planar lipid bilayers reconstituted with ANT1, site-directed mutagenesis and molecular dynamics simulations, we show that the FA anion is first attracted by positively charged arginines or lysines on the matrix side of ANT1 before moving along the positively charged protein–lipid interface and binding to R79, where it is protonated. We show that R79 is also critical for the competitive binding of ANT1 substrates (ADP and ATP) and inhibitors (carboxyatractyloside and bongkrekic acid). The binding sites are well conserved in mitochondrial SLC25 members, suggesting a general mechanism for transporting FA anions across the inner mitochondrial membrane.

## 1. Introduction

The proton gradient Δμ(H^+^) across the inner mitochondrial membrane (IMM) is coupled to ATP production. The adenine nucleotide translocase (ANT, also AAC for ADP/ATP carrier) exchanges cytosolic ADP for matrix ATP to maintain energy production and supply. Dissipation of Δμ(H^+^) without the formation of ATP (uncoupling) is mediated by uncoupling protein 1 (UCP1), which produces heat during non-shivering thermogenesis in the brown adipose tissue of hibernating animals and newborns [1]. Several other proteins of the SLC25 superfamily, including UCP2, UCP3 [2,3] and ANT1 [4,5,6,7], also transport protons, although the exact biological function and significance of the transport are unclear. The H^+^ turnover numbers of UCP1-UCP3 and ANT1 in the presence of arachidonic acid are similar, indicating a similar potency for H^+^ transport [6,8,9,10].

Two mechanistic models have been proposed to account for the protonophoric activity of ANT in the presence of long-chain fatty acid (FA) [11]. In the FA cycling model (Figure 1a) [4,12], protons are transported by a flip–flop of the protonated fatty acids [12], which quickly diffuse through the lipid bilayer to the matrix side [13,14], while a transport of the anionic form of the FA (FA^−^) back to the intermembrane space is assisted by mitochondrial carriers, such as ANT1 and UCPs. An alternative model assumes that the fatty acid is bound in the ANT translocation pathway to support proton translocation through the center of the protein [7]. This model is reminiscent of the “proton buffering” model proposed by Klingenberg et al., for UCP1 [15]. FA is proposed to activate proton leak in either c- or m-state of ANT. However, this model cannot explain the experimentally shown dependence of ANT-mediated proton transport on the FA structure and pH as previously shown [6]. Moreover, the movement of the proton through the positively charged cavity (electrostatic potential about 1 V [16]) is unlikely.

Our previous results imply that an FA^−^ most likely moves through the bilayer at the ANT1—lipid interface [6], and its transport is best described with the FA cycling model (Figure 1a). Fatty acids are weak uncouplers and can themselves transport protons to a certain extent [9]. In this model, the role of the proteins (UCP, ANT) is to assist the reverse transport of the FA anions, which is the rate-limiting step. The reported dependence of membrane conductance on fatty acid saturation would support fatty acid cycling [6]. However, the molecular details of the mechanism by which ANT1 transports FA anions back to the intermembrane space remain unclear. We have used specific recombinant ANT1 mutants and a series of unbiased MD simulations based on the crystallographic structure of ANT1 [17] to analyze the FA^−^ transport pathway. We provide new evidence for our proposed FA^−^ sliding mechanism [6,18]. As the pathway utilizes highly conserved amino acids, it may also be used by other uncoupling proteins to transport FA^−^.

## 2. Results

### 2.1. Identification of the Initial AA^−^ Binding Site on the ANT1 Matrix Side

Based on the electrostatic surface potential analysis of ANT1 and on the inspection of the location of positively charged amino acids on the matrix side (Figure 1b), we identified at least four positively charged lysine and arginine residues (K48, K51, R59 and K62, (Appendix A)) as likely to represent an initial binding site for arachidonic acid (AA) anions. Using site-directed mutagenesis, we produced ANT1 mutants in which these amino acids were substituted with the polar but neutral serine (K48S, K51S, R59S and K62S) and reconstituted them in a planar bilayer membrane [19] to compare their protonophoric activity with that of the wildtype (wt) proteins. This model has the advantages that all components are well defined and purified ANT1 is the only protein in the membrane (Appendix A) [6]. Because the correct refolding of the protein is crucial for the system, we performed a series of control experiments to determine the ADP/ATP exchange rate using either a Mg^2+^-sensitive fluorescence dye Magnesium Green [20] or radioactively labeled substrates ([6] and Appendix A). The ADP/ATP exchange rates of 3.49 ± 0.41 mmol/min/g [20] and 5.53 ± 0.74 mmol/min/g [6] are in good agreement with data in the literature [21,22].

The measurements of the total membrane conductance (G_m_) represent FA^−^ transport due to two reasons: (i) only H^+^ are transported by FA and contribute to the G_m_ (Kreiter et al., 2021); and (ii) the translocation of the FA^−^ is the rate-limiting step in the FA cycling model. G_m_ was determined from current voltage (I–V) characteristics, which were linear in the range from −50 to 50 mV. Appendix A shows further that a significant increase in the current of the ANT1-containing membrane occurs only after the addition of AA and can be inhibited by ATP (see also [23]). Comparison of G_Mutant_ of membranes reconstituted with the mutants with G_ANT1_ for the wt protein in the presence of AA revealed that G_Mutant_ was decreased by 70% in ANT1-K48S, ANT1-R59S and ANT1-K62S but was almost unaffected in ANT1-K51S (Figure 1c and Appendix A). ATP has no inhibitory effect in any mutant except ANT1-K51S (Figure 1d and Appendix A), indicating the loss of coupling between ADP/ATP exchange and AA^−^ trapping. Appendix A provides the absolute values of the total membrane conductance.

To assess whether the ANT1 mutations affect the protein-mediated substrate binding, we measured the time course of the ^3^H-ATP concentration in liposomes in the absence of the protein and in the presence of reconstituted ANT1 or mutant (Appendix A). The relative ratios (k_Mutant_/k_ANT1_) show that mutations had only a slight effect on the ANT1-mediated ADP/ATP exchange rate, k, (Figure 1e, Appendix A), apart from ANT1-R59S, which is involved in attracting AA^−^ but neither in binding the substrates nor in the matrix salt bridge network [16,24].

Molecular dynamics simulations show that the AA^−^ is trapped in the vicinity of these amino acids (Figure 1f and Appendix A). The average distances between the carboxyl carbon atom of the AA anion and R59 or K62 (Figure 1g) are sufficiently short to ensure attractive electrostatic interactions via salt bridges. We also observed partial interactions of AA^−^ with K48, although the separation is slightly larger, while the distance between AA^−^ and K51 was much larger (Figure 1g), consistent with our conductance measurements (Figure 1c). Thus, the unbiased MD simulations also indicate that K51 does not participate in attracting AA^−^ to the protein.

### 2.2. AA^−^ Slides along a Positively Charged Surface at the Protein–Lipid Interface

ANT1 has a considerable positive surface potential at the protein–lipid interface (Figure 1b). The positive potential extends deep into the center of the bilayer membrane and includes the substrate-binding center in the protein interior [16]. We had previously constructed trajectory density maps using a series of short unbiased MD simulations in which the AA^−^ was randomly placed in the vicinity of the ANT1 surface along the positively charged electrostatic potential of ANT1 (see Figure 1b from [18]). The results of MD simulations [18] imply that AA^−^ may slide along the protein–lipid interface to the substrate-binding center of ANT1, which includes at least three charged arginines (R79, R137 and R279) and K22 (Appendix A).

To understand which amino acids participate in the FA sliding pathway, we analyzed the protonophoric activity of the mutants ANT1-R79S, ANT1-K22S, ANT1-R137S and ANT1-R279S, in which arginines and lysine were substituted by a neutral serine (Figure 2a). The conductance of the membranes reconstituted with ANT1-R79S was 30% lower than that of ANT1 (Figure 2a and Appendix A). Although this effect is not very large, additional mutation of the positively charged residues surrounding the R79 region (K22 and R137, Figure 2a) can contribute to the decrease in the proton transport. Positively charged amino acids have been reported to have a role in AA^−^ binding, although in the context of a different transport model [7]. In contrast, we found no effect of mutation of R279S on the membrane conductance, which is different to the results reported in another study [25].

ATP did not inhibit proton transport in mutant ANT-R79S (Figure 2b and Appendix A), again showing the essential role of R79 for ATP binding. The inhibition was less strong for ANT1-K22S, ANT1-R137S and ANT1-R279S (Figure 2b and Appendix A), which may also be involved in binding ATP [26,27,28]. The ADP/ATP exchange rate, measured using ^3^H-ATP in proteoliposomes, was significantly lower only in ANT1-R79S (Figure 2c and Appendix A), confirming the importance of this amino acid in nucleotide exchange [21]. These experimental data suggest that R79 may represent a second crucial binding site for AA^−^ sliding along the ANT1/membrane interface.

We then addressed whether AA^−^ remains bound to the protein at R79 until its protonation or moves further to the cytosolic side of the protein. We mutated K93 to serine (K93S), another positively charged residue that points to the lipids at the membrane/water interface. The mutation led to only slight alterations in membrane conductance (Figure 2a and Appendix A), inhibition by ATP (Figure 2b and Appendix A) and ADP/ATP exchange (Figure 2c and Appendix A). The lack of effect of the K93S mutation may indicate that the AA^−^ will not slide from R79 where it can be protonated since it has access to cytosolic water. On the other side, AA^−^ does not spontaneously exit to the cytosolic side of the lipid bilayer, as expected, due to the large energy barrier for anion transport across bilayers [13,29]. However, other binding possibilities cannot be excluded as ANT1 has a three-fold symmetry.

To establish whether AA^−^ binds to R79, we performed unbiased 1 µs MD simulations. Figure 2d shows the AA anion translocation starting from the matrix side and occurring between transmembrane helices 2 and 3 (TM2 and TM3). The path includes R137, where the AA^−^ binds after 30 ns and stays until 335 ns. It then slides to R79 and K22, where it remains until the end of the simulations (1 µs). The series of snapshots were taken from the MD simulations shown in Appendix A.

The distances between the carboxyl carbon atom of AA anion and the central side chain atoms of K22, R79 and R137 in the MD simulations (Figure 2e) show that AA spontaneously detached after interaction with R137 (evidenced by the distance increase) and attached to R79 (AA-R79 distance decreased) via the interaction with K22 (as visualized by the stepwise decrease in AA-K22 distance, Figure 2e). It is worth noting that no MD simulations of AA^−^ binding to the specific residues of ANT1, including R79, have been reported.

Since cardiolipin is an essential lipid in the inner mitochondrial membrane [24,30,31], we performed the simulations in the presence of three cardiolipin molecules described in the crystal structure of ANT1 [17]. The cardiolipins are located between helices 1–2, 3–4 and 5–6 on the matrix side of the protein (Appendix A). This position is stable in time and is close to the proposed sliding path of FA. Cardiolipin binds tightly to ANT1 and introduces negative charges into the lipid–protein interface [30,32], which affects the AA^−^ sliding pathway.

### 2.3. The AA^−^ Is Protonated in the Hydrated Cavity of ANT1

MD simulations show that the positively charged amino acids R79 and K22 ensure the initial stabilization of the negative headgroup of the AA anion for more than 600 ns (Figure 2e). The average occupancy of the AA^−^ during 1000 ns (Figure 3a) implies that the hydrophobic tail of the AA^−^ can be additionally stabilized at the TH2–TH3 interface with both the hydrophobic surface of the protein and hydrophobic tails of the lipids in the lipid interior.

This conclusion is in agreement with published data showing that AA^−^ is stabilized with its headgroup between helices inside the protein and its tail outside in the lipid bilayer [25]. In this energetically favored position, the AA^−^ has access to the cytosolic aqueous environment for protonation (Figure 3c), and the presence of the negatively charged aspartic acid 134 (D134) in the vicinity of R79 (Figure 3b) can facilitate AA^−^ protonation as described earlier [33]. It is important to emphasize that, even though the sliding mechanism postulates that R137 is involved in a salt bridge network, the protein remains impermeable to water (Figure 3c), which is a strong indication that protein stability and H^+^ transport capability are preserved, in contrast to the case of membrane proteins extracted from phosphocholine detergents, where water leakage inhibits proton transport [34,35].

To test whether the negatively charged D134 facilitates AA^−^ protonation after binding to R79, we substituted D134 with S134 using site-directed mutagenesis. AA^−^ transport was reduced by 60% (Figure 3d and Appendix A), highlighting the important role of the D134 residue in the FA sliding mechanism. The inhibition of ANT1-D134S by ATP was stronger than the inhibition of ANT1 (Figure 3e and Appendix A). The effect can be explained by the reduction in one net negative charge in the substrate binding site of ANT1, which facilitates ATP binding in electrostatic terms. The mutation did not alter ANT1-mediated ADP/ATP exchange (Figure 3f and Appendix A). Our results strongly suggest that protonation of AA^−^ is possible with the support of D134, which is located nearby, as shown by the analysis of MD simulation structures (Figure 3b). They also show that the main nucleotide exchange function of ANT1 protein is conserved, pointing to the conserved protein stability upon mutation.

### 2.4. Protonated Arachidonic Acid Readily Detaches from R79

The exit of protonated AA represents the critical step in the proposed FA sliding mechanism. The literature shows that a close spatial arrangement of positively and negatively charged amino acids promotes the attraction of a proton from water in the cavity [33]. The event is crucial for substrate/H^+^ co-transport in several mitochondrial carriers, such as phosphate and aspartate/glutamate carriers [33,36,37]. MD simulations showed that AA^−^ binds to R79 for at least 600 ns (Figure 2d). To check the effect of protonation of AA^−^ bound to R79, we ran an additional 1 µs simulation of the protonated AA (Figure 4 and Appendix A). We found that AA in the protonated form at the R79 binding site readily detached and remained at the TH2–TH3–lipid interface for nearly 90 ns. It then moved into the lipid bilayer on the cytosolic side, where it stayed until the end of the simulation (1 µs, Appendix A).

### 2.5. R79 Mediates the Competition of AA^−^ and Substrate Transport in ANT1

R79 is important for the binding not only of AA^−^ but also of the substrates ATP and ADP and the ANT1 inhibitors CATR and BKA (Figure 5a,b, Appendix A). We investigated whether R79 mediates the competition between the AA^−^ and ATP, CATR and BKA. Figure 5c and Appendix A show that ADP/ATP exchange is negatively regulated by the presence of an AA in a concentration-dependent manner. The ADP/ATP exchange of ANT1-R79S was generally lower but unaffected by the presence of AA (Figure 5c and Appendix A). The addition of 100 µM ATP changed the membrane conductance for ANT1 depending on the AA concentration in the membrane (Figure 5d and Appendix A). The conductance of the membrane reconstituted with the ANT1-R79S mutant was not altered (Figure 5d and Appendix A). Similarly, AA content had a strong effect on the inhibition of ANT1 by CATR and BKA but far less effect on the inhibition of ANT1-R79S (Figure 5e,f and Appendix A). The experiments show that ANT-mediated ADP/ATP exchange and AA^−^ transport are interdependent and negatively regulate each other by competitive binding at R79.

## 3. Discussion

We report a new molecular mechanism by which ANT1 transports FA anions across the inner mitochondrial membrane, which we have named the “FA sliding mechanism”. The sliding mechanism explains how an FA anion moves to the cytosolic monolayer of the IMM and completes the catalytic cycle of H^+^-transport in the FA cycling model [12,38]. The initial idea on mitochondrial uncoupling was that it counteracts the generation of reactive oxygen species, as proposed for UCP2 and UCP3 [11]. However, the transport of FA anions to the cytosol driven by proton gradient is also of great physiological significance as it prevents the accumulation of free FAs in the mitochondrial matrix. Excessive amounts of FAs induce lipotoxicity and are involved in the pathogenesis of obesity, diabetes, ischemia and degenerative diseases [39,40,41,42,43].

The similarity of the H^+^ transport characteristics between ANT1 and UCPs implied that the FA^−^ may be trapped by positively charged amino acids on the m-side of ANT1 (Figure 6), as proposed for UCP2 [44,45]. Our results contradict previous reports that FAs bind to the protein exclusively from the cytosolic side (c-side) [46] and are supported by the observation that an AA^−^ is readily protonated on the cytosolic side due to the pH gradient. Data from experiments [13,14] and MD simulations [29] show that the neutral form can diffuse across the IMM in millisecond range.

After an FA^−^ binds to the matrix side, it may follow the positively charged surface of ANT1 (Figure 6), as proposed earlier [6]. Longer and more unsaturated FAs diffuse faster through the lipid bilayer, and their pK_a_ in lipid bilayers is closer to the physiological pH [47]. pH is crucial to accelerate protonation/deprotonation at the lipid–water interface [48]. The FA^−^ translocates along the protein–lipid interface such that the negatively charged FA carboxylic group is in close contact with the positive-charged amino acids of the protein surface, while the hydrophobic chain moves through the hydrocarbon layer of the lipid membrane adjacent to the protein. The effect is analogous to the “credit card swipe” model proposed for lipid scramblases and flippases [49,50,51,52,53,54,55] based on the crystallographic structure of the calcium-activated chloride channel (CaCC/TMEM16) and the lysosomal integral membrane protein 2 (LIMP2), which function as a lipid scramblase or an FA translocase [50,56,57]. Both proteins contain a membrane-spanning hydrophilic groove at the protein–lipid interface, in which the phospholipid/FA head is inserted and through which it is transported, while the hydrophobic tail remains inside the lipid bilayer. The presence of a hydrophobic pocket in the protein, in which longer and more unsaturated FA would reside for longer, would lead to faster diffusion, as discussed above, but the resolved structures of ANT1 show no such hydrophobic pocket, ruling out this possibility [17,58]. Furthermore, lipid transfer through the groove is coupled to the leakage of lipid-conjugated ions, such as Ca^2+^, making a similar mechanism unsuitable to account for FA^−^ transport across the inner mitochondrial membrane. Previous work with modified lipid heads of various sizes suggested that lipid scrambling also occurs on the protein–lipid interface outside the groove of CaCC, excluding the transport of lipid-conjugated ions [49,59]. Similarly, our MD simulations show that the FA^−^ moves along the lipid–protein interface from the positively charged cloud on the matrix side to R79 between TH2 and TH3, largely driven by the positive electrostatic surface potential of ANT1. In this way, FA^−^ transport is not associated with any unwanted ion leakage across the IMM and mitochondrial function is not compromised. The FA anion is more likely to be translocated along the protein–lipid interface than to reside inside the protein, where long hydrophobic aliphatic chains would be exposed to water, which is energetically unfavorable.

Next, we identified a second FA^−^ binding site—R79—in the substrate binding site of ANT1 (Figure 6) [60,61]. This is of particular interest because the substrates (ADP and ATP) and the inhibitors (CATR and BKA) use R79 as a common binding target [16,17,26,62,63,64]. Not only does ADP/ATP exchange negatively regulate FA transport but FA^−^ and the substrates (CATR and BKA) also compete for R79 (Figure 5). In the presence of FA, the ADP/ATP exchange of ANT1 is diminished, but the effect is not shown by the mutant ANT1-R79S. We concluded that the FA anion does not displace the ADP or ATP but only decreases the binding affinity for these substrates as its binding blocks the R79 residue. Binding disfavors attractive electrostatics by reducing the positive charge of the positive R79, which would otherwise be accessible to ATP, leading to decreased ATP binding. Interestingly, other groups postulated the existence of a similar binding target in the central cavity of the carriers UCP1 and ANT1, but the precise amino acid has remained unknown [15,46,65]. The binding of FAs to the center of protein was also described in [25]; however, the FA binding to R279 was only inferred by MD simulations and experimental verification of the MD results is missing. We have clearly shown that (i) the mutation of R279 does not alter FA^−^/H^+^ transport; and (ii) competition between FA and ANT1-specific substrates ATP, CATR and BKA is mediated by R79, which is the common binding target for FA and all other substrates.

The role of the FA is different in FA cofactor and FA cycling models. In the first model, FAs bind to ANT1 or UCP1 to enhance H^+^ translocation across the protein by breaking the tight salt bridge network of the protein on the matrix side, which is energetically costly. Both protonation and deprotonation of the FA are thought to occur inside the protein cavity [7]. In contrast, the FA cycling mechanism assumes that protonation/deprotonation of the FA occurs only at the lipid–water interface [12].

Additionally, we have found strong evidence for a novel and critical third step in the FA sliding model. Instead of being transported further to the cytosolic side, the FA^−^ is protonated while bound to R79 (Figure 6). R79 attracts the FA^−^ carboxylic head into the hydrated protein cavity and the spatially close D134 mediates proton transfer from the hydrated cavity to the FA. Several mitochondrial carriers that co-transport H^+^ and substrates, such as the phosphate carrier [36], the aspartate/glutamate carrier [37], the oxodicarboxylate carrier [66] and the GTP/GDP carrier [33], may use a similar protonation/deprotonation mechanism. Small amounts of H^+^ are moved together with the ADP/ATP exchange, but proton exchange does not always occur [67,68]. In the final step, the neutral FAs are almost immediately released into the lipid bilayer (Figure 6, Appendix A) and the catalytic cycle can start again. Thus, ANT1 is a carrier for FA^−^ that fits the established view that all functionally characterized members of the mitochondrial carrier family SLC25 are anion carriers [69,70].

The proposed FA^−^ translocation mechanism mediated by ANT1 is energetically more favorable than the FA cofactor or the FA shuttling model proposed for ANT1 and UCP1 [46,65]. Our direct unbiased MD simulations show a spontaneous translocation of FA (FA sliding) across the membrane at the microsecond timescale, which suggests relatively low translocation barriers (below 5 kcal mol^−1^). Furthermore, if the FA promotes H^+^ transport through the protein center by either FA flip–flop or as a co-factor, an FA carboxylic group carrying an H^+^ has to cross at least one salt bridge network that closes ANT1 either to the cytosolic or to the matrix side [69]. As part of the alternating access mechanism [58], salt bridge network must function properly and ANT1 not undergo a spontaneous conformational change in the absence of ADP or ATP [24,27], which presumably requires at least 10 kcal mol^−1^ [71]. The energy released by the binding of the singly charged FA^−^ or H^+^ to the cavity is predictably too small to break the salt bridge network, in contrast to the large amount of energy released when the multiple negatively charged ATPs and ADPs bind to the positively charged bottom of the protein cavity. In addition, MD simulations of ANT1 and UCP2 in DOPC bilayers show that water, including H_3_O^+^, is hindered by the cytosolic or matrix salt bridge network and cannot leak through the protein [24,45]. A substantially different model proposes that UCP2 forms tetramers, mediating FA-activated H^+^ transport [72]. The authors suggest that ANT1 tends to form dimers and tetramers to facilitate proton transfer, although this has not been studied in detail [73]. We show that FA can be transported through the ANT1 monomer, as shown for ATP/ADP exchange [74,75]. Oligomerization is not required for FA anion transport, although we cannot exclude the possibility that it favors FA^−^ transport over nucleotide transport to some extent.

The described ANT1-mediated FA^−^ transport involves amino acids that are well conserved among SLC25 members [76]. We suggest that the mechanism is general and also applies to other mitochondrial carriers that exhibit protonophoric activity in the presence of FA, such as the uncoupling proteins, the dicarboxylate and the aspartate/glutamate carriers [11,77,78]. Indeed, the activation and inhibition pattern of UCP1-3 is consistent with the FA sliding model [3,9,10]. R79, R137 and D134 included in FA sliding are conserved and have equivalents in UCP1-3. The precise mechanism of FA^−^ transport might differ, and the mechanistic details of each mitochondrial carrier await elucidation at the molecular level.

## 4. Materials and Methods

### 4.1. Chemicals

Sodium sulfate (Na_2_SO_4_), 2-(*N*-morpholino)ethanesulfonic acid (MES), tris(hydroxymethyl)-aminomethane (Tris), sodium dodecyl sulfate (SDS) ethylene glycol-bis(β-aminoethyl ether)-*N*,*N*,*N*′,*N*′-tetraacetic acid (EGTA) were purchased from Carl Roth GmbH & Co. K.G. (Karlsruhe, Germany). Hexane, hexadecane, dimethyl sulfoxide (DMSO), purine nucleotides adenosine tri-, di- and guanosine triphosphate (ATP, ADP, GTP), agarose, carboxyatractyloside (CATR) and bongkrekic acid (BKA) were purchased from Sigma-Aldrich Co. (Vienna, Austria). 1,2-dioleoyl-sn-glycero-3-phosphocholine (DOPC), 1,2-dioleoyl-sn-glycero-3-phosphoethanolamine (DOPE) and cardiolipin (CL) came from Avanti Polar Lipids Inc (Alabaster AL, USA) and chloroform from Sanova Pharma GesmbH (Austria). Arachidonic acid (AA) was purchased from Larodan (Biozol, Eching, Germany).

### 4.2. Cloning, Site-Directed Mutagenesis, Isolation and Reconstitution of Murine ANT1

Cloning, isolation and reconstitution of murine ANT1 and ANT1 mutants followed an established protocol [20]. In brief, for protein expression, we used E. coli strain Rosetta (DE3; Novagen). To isolate inclusion bodies, we centrifuged cells disrupted by high-pressure cell disrupter *One Shot* (Constant Systems Limited, Daventry, UK) at 1 kbar. For reconstitution, protein from inclusion bodies was first solubilized in 100 mM Tris at pH 7.5, 5 mM EDTA, 10% glycerin (TE/G-buffer) containing 2% sodium lauryl sulfate and 1 mM DTT. Then, it was gradually mixed with the membrane-forming lipids (DOPC, DOPE and CL; 45:45:10 mol%) dissolved in TE/G-buffer containing 1.3% Triton X-114, 0.3% n-octylpolyoxyethylene, 1 mM DTT and 2 mM GTP. After several dialysis steps, the mixture was dialyzed three times against assay buffer (50 mM Na_2_SO_4_, 10 mM MES, 10 mM Tris, 0.6 mM EGTA at pH 7.35). To eliminate aggregated and unfolded proteins, the dialysate was centrifuged and run through a hydroxyapatite-containing column (Bio-Rad, Munich, Germany). Non-ionic detergents were removed using Bio-Beads SM-2 (Bio-Rad). To ensure the correct folding of the used proteins, we performed control experiments described previously [6]. They show that the ATP binding constant in the absence of AAs is in the μM region, which is similar to the literature values.

To generate ANT1 mutants, we carried out site-directed mutagenesis of lysine 48, 51, 62, 93, arginine 59, 79, 137, 279 and aspartic acid 134 to serine using the Q5 Site-Directed Mutagenesis Kit (New England Biolabs, Graz, Austria). The sequences were verified by Sanger sequencing. Mutants were always refolded in parallel with wildtypes to minimize artefacts. Critical mutants such as R59S and R79S were refolded at least twice to rule out any possible side effects of the refolding.

The protein concentration of the proteoliposomes was measured with the Micro BCA^TM^ Protein Assay Kit (Thermo Fisher Scientific, Vienna, Austria; Prod. #23235). SDS-PAGE and silver staining verified protein purity (Appendix A). Furthermore, we have performed the control measurements of ADP/ATP exchange (Appendix A), activation by free fatty acids and inhibition by 2 mM ATP of both the mutant and the parallel refolded wildtype (Appendix A) to ensure the protein functionality.

### 4.3. Exchange Rate Measurements of ANT1

ANT1-mediated exchange of ATP/ADP was measured using ^3^H-ATP (Perkin Elmer Prod.# NET420250UC, Waltham, MA, USA) [20]. The standard protocol was slightly modified in that ^3^H-ATP was initially present in the proteoliposomes to measure the release of the radionucleotide over time (Appendix A).

### 4.4. Planar Bilayer Membrane Formation and Conductance Measurements

Planar lipid bilayers were formed at the top of the dispensable plastic pipette as described previously [19]. AA was added directly to the lipid phase before membrane formation. We verified proper membrane formation by measuring the membrane capacitance (C = 0.72 ± 0.05 µF/cm^2^). C was independent of the presence of protein, AA and inhibitor. Current voltage (I-U) recordings were performed with a patch clamp amplifier (EPC 10USB, Harvard Bioscience, Holliston, MA, USA). Total membrane conductance (G_m_) at 0 mV was derived from the slope of a linear fit of the experimental data at applied voltages from −50 mV to + 50 mV (Appendix A). ATP in buffer solution (pH = 7.34) and the ANT-specific inhibitors BKA and CATR (in DMSO) were added to the buffer solution before forming bilayer membranes. Concentrations were 2 mM ATP in Figure 1, Figure 2 and Figure 3 and 100 µM ATP, 10 µM CATR or 10 µM BKA in Figure 5 and Appendix A. The volume of the added inhibitors in DMSO did not exceed 10 µL and did not alter total membrane conductance as previously shown [6]. The concentrations of each substrate are indicated in the figures. Membrane conductance expressed in relative units was calculated as previously described [10].

### 4.5. Molecular Dynamics Simulations

We performed all-atom molecular dynamics (MD) simulations of ANT1 in a 1,2-dioleoyl-sn-glycero-3-phosphocholine (DOPC) bilayer with the addition of a single AA molecule (20:4), either in anionic or protonated form. Residues 1 and 294–297 missing from the crystal structure of bovine ANT1 open to cytosolic state (c-state, PDB code 1okc) [17] without CATR were added using Modeller9 [79] and implemented into the DOPC bilayer using CHARMM-GUI [80,81,82]. We kept all 7 crystallographically determined positions of lipids and added additional 143 DOPC lipids using charmmgui protocol (146 DOPC molecules in total, 73 per leaflet). Additionally, three cardiolipin (CL) molecules were added in the system to surround the protein at the crystallographically determined positions [17] (Appendix A).

Bovine ANT1 has 95% homology to murine ANT1, which we used as a recombinant protein. The mechanism of ATP/ADP transport involves a conformational change of the protein from the c-state to the matrix open state (m-state) [71]. Since the m-state structure is less stable and short-lived, we assumed that the conformational change and m-state structure are not relevant for the transport of FA anion and performed all simulations only with the ANT1 in c-state.

In total, four systems were simulated: (I) 100 ns of ANT1—electrostatic potential of protein, (II) 200 ns of ANT1 with AA^−^ at matrix side in vicinity of positive cloud (K51, R59, K48 and K62)—initial attraction of AA^−^ to protein, (III) 1000 ns of ANT1 with AA^−^—the initial structure is a snapshot from the system II—sliding along protein, (IV) 1000 ns of ANT1 with AA—the initial structure is protonated AA^−^ at t = 1000 ns from the system III.

Initial simulation box contained ANT1 protein (with a total charge of +19), 73 DOPC molecules per leaflet (146 per system), ~11,500 water molecules, 3 CL molecules, AA^−^/AA if needed and the necessary number of Cl anions to neutralize the net charge. Each system was first minimized and equilibrated in six steps using the CHARMM-GUI protocol [83], and then simulated without any restraints with a time step of 2 fs in a periodic rectangular box of 7.9 nm × 7.9 nm × 9.4 nm using the isobaric–isothermal ensemble (NPT) and periodic boundary conditions in all directions at T = 310 K, maintained via a Nosé–Hoover thermostat [84] independently for the DOPC, water/ions and protein subsystems with a coupling constant of 1.0 ps. The pressure was set to 1.013 bar and controlled with a semi-isotropic Parrinello–Rahman barostat [85] with a time constant for pressure coupling of 5 ps. We used the particle mesh Ewald (PME) method [86] to calculate long-range electrostatics. Real-space Coulomb interactions were cut off at 1.2 nm using a Fourier spacing of 0.12 nm and a Verlet cut-off scheme. Analysis of an electrostatic potential map of ANT1 revealed a tentative translocation pathway of charged AA species, namely the path following positive electrostatic potential (see Figure 1b) corresponding to regions abundant in arginine and lysine residues. All simulated systems were described by the CHARMM36m force field [87]. Electrostatic potential is computed using PMEpot plugin in VMD for each frame of the whole system (protein, water, lipids) and then averaged over the entire trajectory. Particle mesh Ewald (PME) algorithm approximates point charges using spherical Gaussians with sharpness controlled by the Ewald factor; electrostatic potential is explicitly calculated solving the Poisson equation over all atoms of the system with three dimensional grid (48 × 48 × 48) and Ewald factor of 0.25 at T = 300 K [88,89]. All simulations were run with the GROMACS 5.1.4 software package [90] and visualized with the VMD molecular graphics program [91].

### 4.6. Statistics

Data from the electrophysiological measurements are displayed as mean ± SD of at least three independent experiments. Each experimentally derived result was the mean membrane conductance from a minimum of two formed bilayer membranes. The recombinant proteins from at least two refoldings were usually used to ensure the reproducibility of the experiments. Statistical significance was calculated using an unpaired *t*-test. Significance values (*p*-values) are visualized in the figures as the following: ns—not significant; *—*p* < 0.05; **—*p* < 0.01; ***—*p* < 0.001.

## Figures and Tables

**Figure 1 ijms-24-13701-f001:**
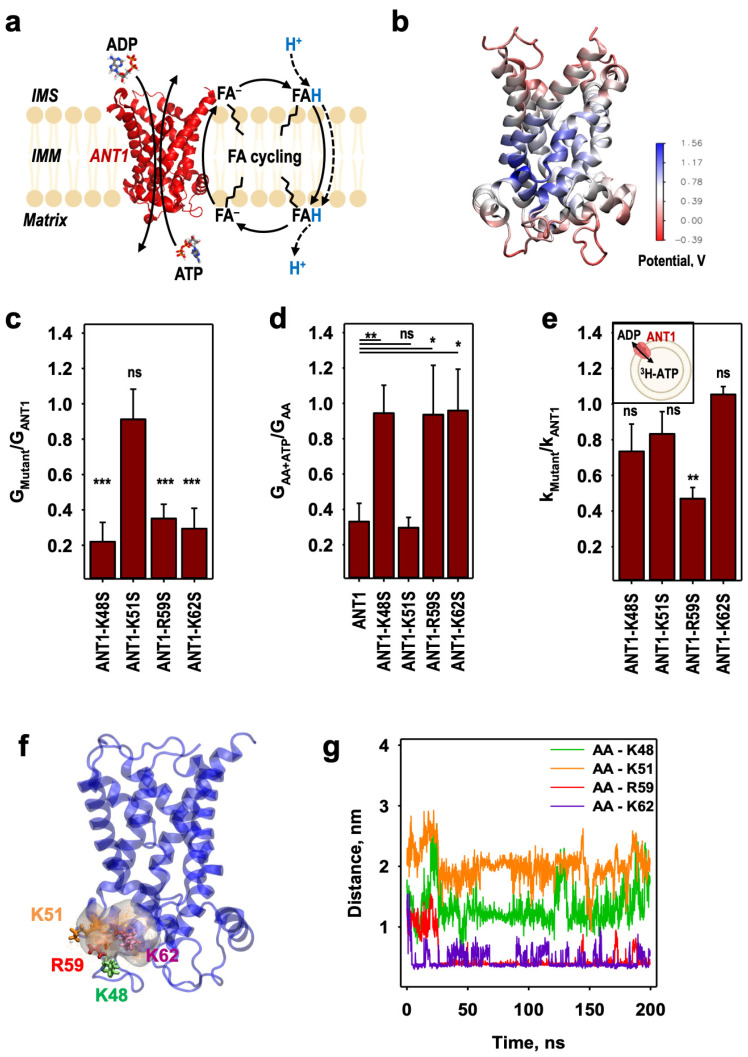
The fatty acid anion (FA^−^) is attracted to arginine and lysine residues at the matrix side of ANT1. (**a**) The FA cycling model, in which ANT1 is proposed to facilitate the transport of FA^−^ across the inner mitochondrial membrane (IMM) from the mitochondrial matrix to the intermembrane space (IMS). (**b**) The calculated electric surface potential of ANT1 (PDB:1OKC). (**c**) The ratio of the total membrane conductance, G_Mutant_, of the membranes reconstituted with ANT1 mutants to G_ANT1_ of the membranes reconstituted with ANT1 in the presence of arachidonic acid (AA) in the membrane. (**d**) The ratio of the total membrane conductance, G_AA+ATP_, in the presence of AA and 2 mM ATP to the total membrane conductance, G_AA_, in the presence of AA only, calculated for the bilayer membranes reconstituted either with ANT1 or ANT1 mutants. (**e**) The ratio of the ADP/ATP exchange rate, k_Mutant,_ of the ANT1 mutants relative to the exchange rate (k_ANT1_) of ANT1 measured with ^3^H-ATP. (**f**) A number density map of the AA anion at the matrix side of ANT1 determined by unbiased molecular dynamics simulations. Amino acids K48 (green), K51 (orange), R59 (red) and K62 (purple) are displayed in licorice. (**g**) Distances between the carboxyl carbon atom of AA anion and central side chain atoms of K48, K51, R59 and K62. In the experiments, the lipid concentration was 1 mg/mL (**e**) or 1.5 mg/mL (**c**,**d**). Protein concentration was 8.5–9 µg/(mg of lipid) (**e**) or 4 µg/(mg of lipid) (**c**,**d**). Membranes were made of DOPC: DOPE: CL (45:45:10 mol%) (**c**–**e**) and reconstituted with 15 mol% AA (**c**,**d**). Buffer solution contained 50 mM Na_2_SO_4_, 10 mM Tris, 10 mM MES and 0.6 mM EGTA at pH = 7.34. Temperature was 296 K (**e**) or 306 K (**c**,**d**). Data are the mean ± SD of at least three independent experiments. *p*-values: ns—not significant; *—*p* < 0.05; **—*p* < 0.01; ***—*p* < 0.001.

**Figure 2 ijms-24-13701-f002:**
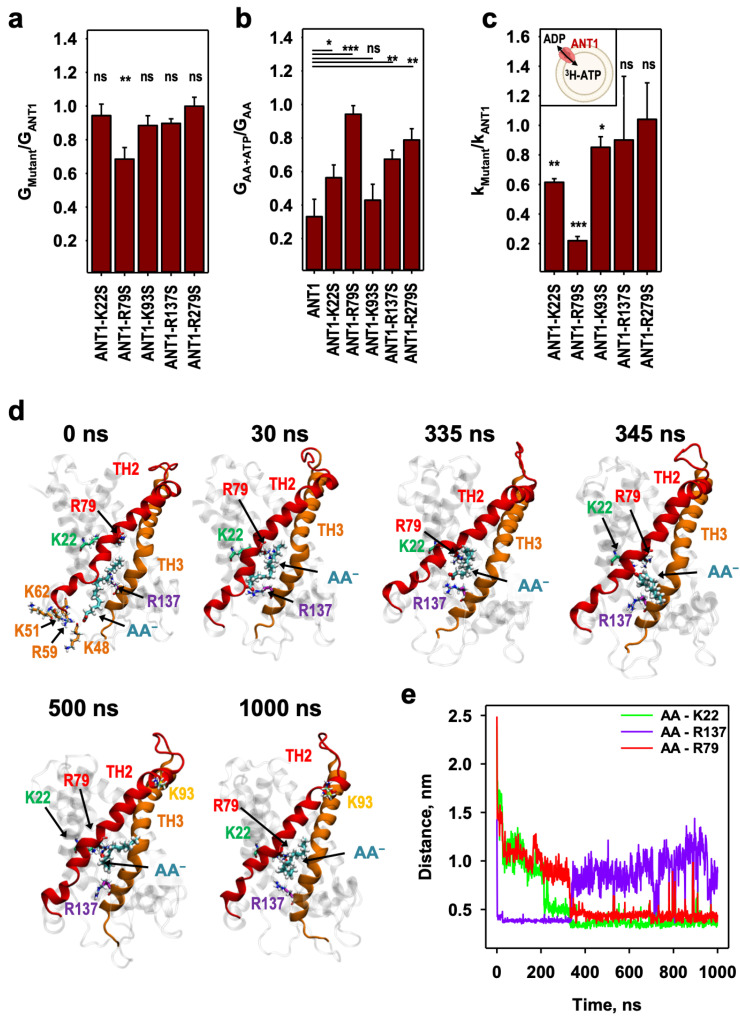
The involvement of different amino acids in the movement of fatty acid anion (AA) to R79. (**a**) The ratio of the total conductance, G_Mutant_, of membranes reconstituted with ANT1 mutants to the G_ANT1_ of the membranes reconstituted with ANT1, measured in the presence of arachidonic acid (AA). (**b**) The ratio of the total membrane conductance, G_AA+ATP_, in the presence of AA and ATP to the G_AA_, in the presence of AA only, calculated for the bilayer membranes reconstituted either with ANT1 or ANT1 mutants. (**c**) The ratio of the ADP/ATP exchange rate, k_Mutant_, of the ANT1 mutants (ANT1-K22S, ANT1-R79S, ANT1-K93S, ANT1-R137S and ANT1-R279S) to the exchange rate, k_ANT1_, of ANT1 measured with ^3^H-ATP. (**d**) A series of MD snapshots of the AA anion translocation between transmembrane helices 2 and 3 starting from the matrix side. Snapshots are extracted from MD simulations shown in Appendix A. Images were taken at t = 0, 30, 335, 345, 500 and 1000 ns. The FA anion (cyan), K22 (green), R79 (red), R137 (purple), K93 (yellow) and K62, K51, R59, K48 (all in orange) are displayed in the licorice representation in the structure of ANT1. (**e**) Distances between the AA anion carboxyl carbon atom and central side chain atoms of K22, R79 and R137 extracted from the MD simulations in [24] (**d**). Experimental conditions are the same as in Figure 1. *p*-values: ns—not significant; *—*p* < 0.05; **—*p* < 0.01; ***—*p* < 0.001.

**Figure 3 ijms-24-13701-f003:**
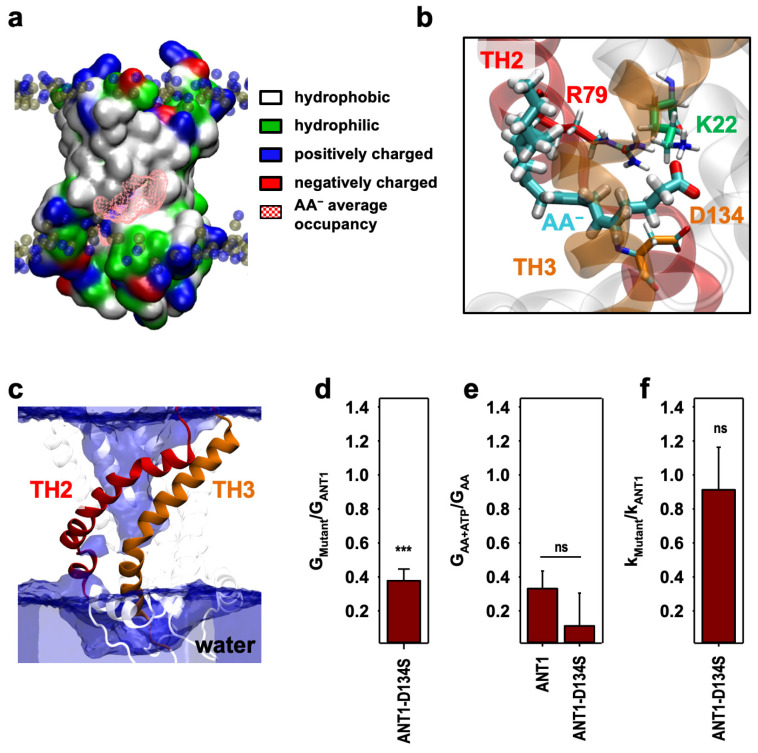
The arachidonic acid anion (AA^−^) protonation at R79 is assisted by D134. (**a**) The average volume of AA^−^ in the 1 µs MD simulation presented in pink wireframe at the surface of ANT1. Amino acids are colored according to their side chain classification (charged, hydrophobic or hydrophilic). (**b**) Position of the AA^−^ carboxylic head inserted between the transmembrane helix (TH) 2 and 3 while bound to R79 with D134 in the vicinity. AA^−^ (cyan), R79 (red), K22 (green) and D134 (orange) are displayed in licorice, whereas TH 2 and 3 are in red and orange, respectively. (**c**) Averaged volume of water (blue surface) within the structure of ANT1 based on the 1 µs simulation in Figure 2. (**d**) The ratio of the total membrane conductance, G_Mutant_, of ANT1-D134S to the G_ANT1_ of ANT1, measured in the presence of arachidonic acid (AA) in the membrane. (**e**) The ratio of the total membrane conductance, G_AA+ATP_, in the presence of AA and ATP to the G_AA_ in the presence of AA only, calculated for the bilayer membranes reconstituted with either ANT1 or ANT1-D134S. (**f**) The ratio of the ADP/ATP exchange rate, k_Mutant_, of the ANT1-D134S to the k_ANT1_ of ANT1, measured with ^3^H-ATP. The experimental conditions are the same as in Figure 1. *p*-values: ns—not significant; ***—*p* < 0.001.

**Figure 4 ijms-24-13701-f004:**
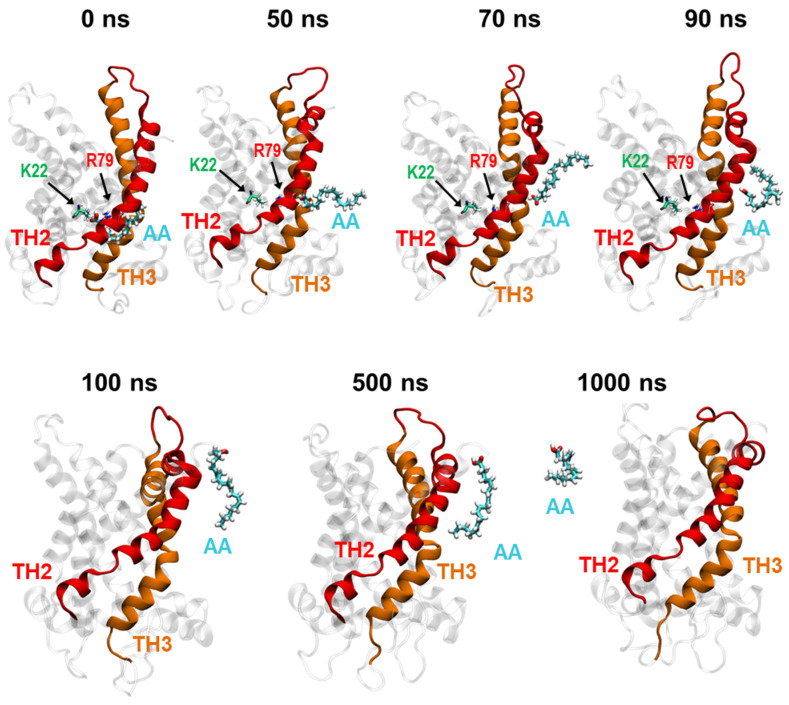
Protonated AA readily detaches from the binding site at R79. A series of MD snapshots of the protonated AA detaching from R79 (red) of ANT1 and its propagation into the lipid bilayer. Snapshots are extracted from MD simulations shown in Appendix A. Images were taken at t = 0, 50, 70, 90, 100, 500 and 1000 ns. The protonated AA (cyan) is displayed in licorice, and transmembrane helices TH2 and TH3 are colored in red and orange, respectively.

**Figure 5 ijms-24-13701-f005:**
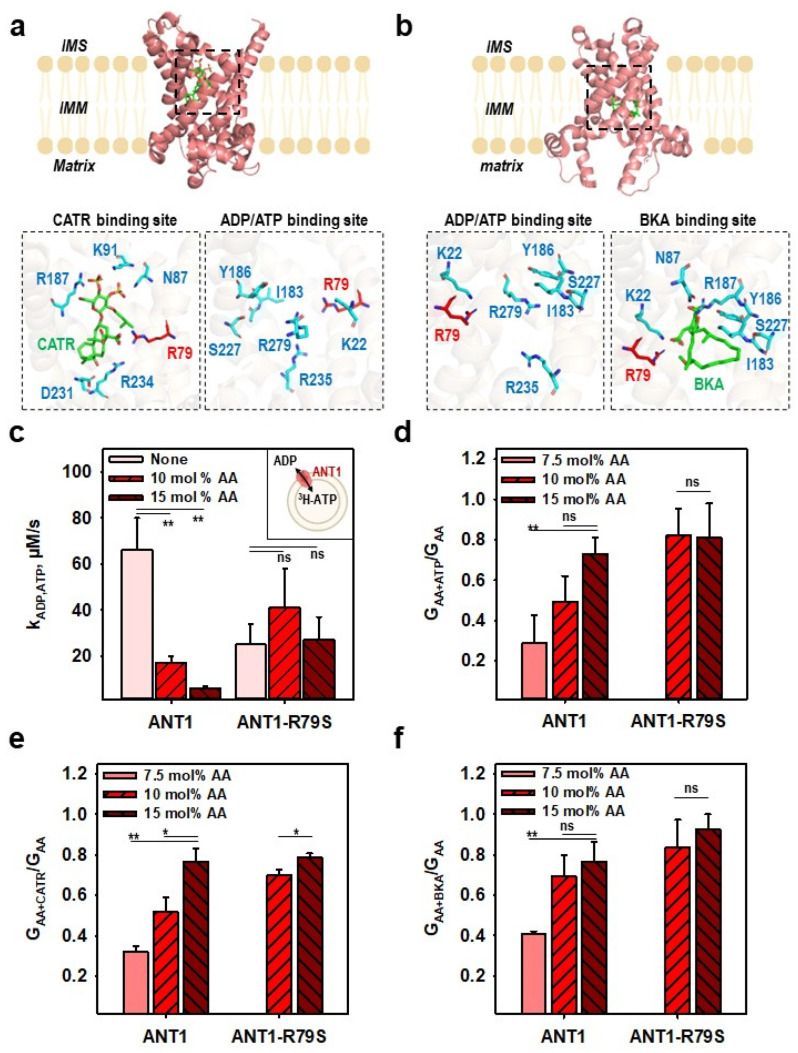
Competition between the arachidonic acid anion (AA^−^) and the ANT-specific substrates binding at R79 in ANT1. (**a**,**b**) Crystallographic structures of ANT1 in complex with carboxyatractyloside (CATR; PDB:1OKC, (**a**)) or bongkrekic acid (BKA; PDB:6GCI, (**b**)). Amino acids involved in the binding of CATR (**a**), BKA (**b**) and ATP or ADP (**a**,**b**) are displayed in licorice in cyan, R79 in licorice in red. The inhibitors CATR and BKA are displayed in licorice in green. Abbreviations are similar to Figure 1. (**c**) ADP/ATP exchange rates (k_ADP, ATP_) of ANT1 or ANT1-R79S in the presence (10 and 15 mol%) and absence of arachidonic acid (AA) in the membrane measured with ^3^H-ATP (inset). (**d**–**f**) The ratio of the total conductance of the membranes reconstituted with ANT1 or ANT1-R79S in the presence of AA and 100 µM ATP (G_AA+ATP,_ (**d**)), or 10 µM CATR (G_AA+CATR,_ (**e**)) or 10 µM BKA (G_AA+BKA,_ (**f**)) to the total membrane conductance in the presence of AA only (G_AA_). In all experiments, the lipid concentration was 4 mg/mL (**c**) or 1.5 mg/mL (**d**–**f**) and the protein concentration was 4 µg/(mg of lipid) (**c**–**f**). Membranes were made of DOPC: DOPE: CL (45:45:10 mol%) and reconstituted with AA in concentrations indicated in the figures. Buffer solution contained 50 mM Na_2_SO_4_, 10 mM Tris, 10 mM MES and 0.6 mM EGTA at pH = 7.34 and T = 296 K (**c**) or T = 306 K (**d**–**f**). Data are the mean ± SD of at least three independent experiments. *p*-values: ns—not significant; *—*p* < 0.05; **—*p* < 0.01.

**Figure 6 ijms-24-13701-f006:**
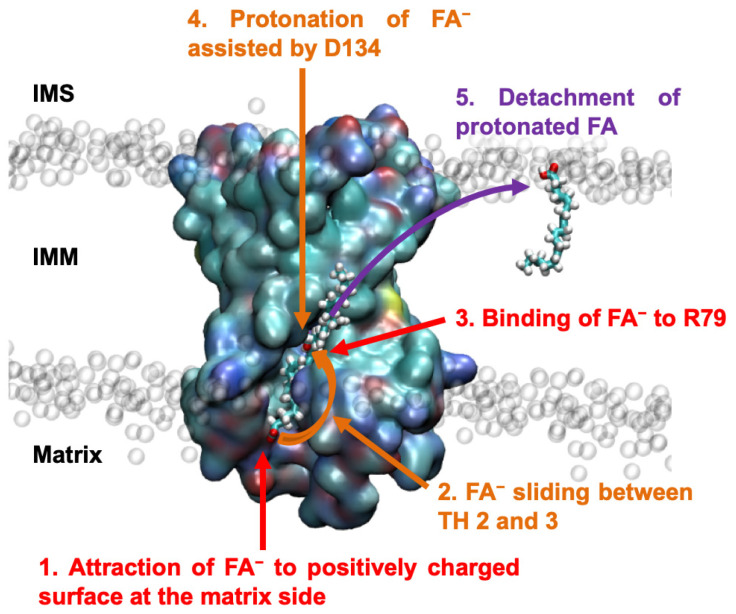
The proposed mechanism of FA anion (FA^−^) transport by ANT1. The FA anion is initially attracted to the positively charged surface at the matrix side of ANT1 around R59 (1). From this binding site, it slides between transmembrane helices TH2 and TH3 to R79 (2). The FA anion subsequently binds to R79 (3), where it is protonated by water in the hydrated cavity assisted by D134 (4). The protonated FA finally detaches from the protein and moves into the lipid bilayer, freely diffusing to the cytosolic side and completing the H^+^ catalytic cycle (5).

## Data Availability

The datasets generated and/or analyzed during this study are available from the corresponding authors on reasonable request.

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
