# Peer review of "FA Sliding as the Mechanism for the ANT1-Mediated Fatty Acid Anion Transport in Lipid Bilayers"

_ijms, 2023, doi:10.3390/ijms241813701_

Round 1

Reviewer 1 Report

The manuscript by Kreiter et al. described experiments and computations that they conducted to determine the molecular mechanism for ANT1 mediated fatty acid anion (FA-) transport in a lipid bilayer. While the text was clearly written and easy to understand, and figures were carefully drew to present computational models and experimental data, there are many aspects that must be improved to sustain the potentially impactful conclusions that the authors reached.

Introduction could be improved as well as other sections. Although the manuscript focuses on proving the FA-cycling model that they and others proposed previous, the H+ buffering model need to be described with more details as well as the existing evidence supporting this competing model, citing additional references if necessary. Then, whether the new data support or refute this model must be mentioned in Results and discussed in Discussion. 

Research design. The main approach has two branches. Experimentally the total conductance of ANT1 protein in lipid bilayer, and the conductance of FA- (AA-) through the protein-lipid interface as the FA-cycling model suggested were measured as well as the ADP/ATP exchange rate. Computationally MD simulations were used to calculate the density map of AA- in a system containing an ANT1 protein in a lipid bilayer surrounded by aqueous solution during the AA- binding, translocation and release phases. In addition, an electric surface potential map was calculated to suggest potential binding sites and translocation path for AA-. While the methods of MD simulations were described with details, the methods of electric potential calculation and some experiments were not. For example, it is unknown whether the electric potential map was calculated for the ANT1 structure in the lipid bilayer. If not, the map may not be relevant to the protein-lipid interface that was hypothesized for the AA- flop from the mitochondrial matrix to the inter membrane space. It is also unknown how the conductance of AA- through the protein in the lipid bilayer was measured or derived from the total conductance measurement. Importantly statistical analyses for the experimental measurable's and derivatives were mostly not described. This raises a question about the significance of the difference between a pair of data (e.g., that obtained from the wild type and a mutant ANT1), which were described in text only by qualitative terms, and in figures only by means and SD's.        

Most critically the experimental design and results' presentation almost completely ignored the contribution of lipids in the lipid bilayer where the ANT1 protein was embedded and functioned. This is a major deficiency of the manuscript that was tried to prove a model from an organic ion to go though a milieu that constitutes not only a protein but also a lipid bilayer, and an aqueous solution whose contribution was also largely uncounted in the experiments and computations. Related to this point, have any structures of ANT1 or other UCPs been determined since 2003 with the protein in a lipid bilayer? If so, would a such structure be a better starting point for computational modeling and molecular simulations?

There are other aspects of the manuscript that could be improved.

Can any MD simulations be done with any ANT1 mutants? If so, are the results different from the wild type protein? Can any MD simulations be done with a DOPC:DOPE:CL lipid bilayer to match the bilayer used in experiments?

Please add results from statistical analyses for all bar and data point-fit curve graphs such as 95% confidance intervals or p-values.

The schematic inserted in Fig. 1e and other figures needs a description in the legend or be referred to in Methods. ANT1-K93S was not mentioned in text. Please add K93 to a structure in Fig. 2d.

Fig. S1a, please show the entire gel so the protein purity and integrity can be evaluated. In Fig S1c, the y-axis label cannot be seen entirely. Fig. S2aShow the conductances that fit the I-V data.   

First paragraph in page 6. How would MD simulation of AA- binding to the protein show the binding affinity to a residue or residues? How would cardiolipin affect the AA- sliding path? Would the authors do MD simulations without cardiolipin to suggest a potential effect?

Line 231, 'conserved' or 'preserved'? Line 455-456, describe the concentration of added inhibitors instead of the volume.  

Fine.

Reviewer 2 Report

This manuscript investigates the possibility of FA sliding as a mechanism for ANT1 mediated fatty acid anion transport. Three possible mechanism are proposed: sliding, cycling and buffering. In the discussion is another model, a shuttling model, mentioned. The authors have not worked out the advantages or disadvantage of each of these models relative to the preferred sliding model.

FAH (used once), abbreviation not necessary

FA- (used a few times), not defined.

The authors should provide some information why arachidonic acid was used. Would their proposed mechanism also work with other fatty acids? I am also wondering, whether the length of the fatty acid may have an impact on the proposed mechanism.

 In line 46 the authors write “while FA- transport back to the intermembrane space is assisted by protein”. Please clarify and maybe explain, whether the mentioned protein is ANT or something else.

Line 53: more “proteins”, that are not defined; please clarify, because it is important to understand the model.

Line 185: protein residues; my guess is that amino acids are meant.

Figure 1f: shows some of the lysine and arginine residues, that have been mutated. For complete understanding, is it possible to add the position of all mutated amino acids? This could be a done either in figure 1f or as a new supplemental figure.

Line 186: the authors write “We performed the simulations in the presence of cardiolipin, an essential lipid in the inner mitochondrial membrane”, but any presentation of these results or the details for the protocol is missing.

Line 235: References are missing to support the statement that the “literature shows …”.

Bar graphs in figures 1 (c,d,e), 2 (a,b,c), 3 (d,e,f), 5 (c,d,e,f) and the supplemental figures (1, 2, 5, 6) show error bars, but no significance. Is it possible to establish whether differences that are shown in these graphs reach significance? It may change how some of the presented data are interpreted.  

 Please check: The use of “the” where a/an would be more appropriate.

Please check: The use of “the” where a/an would be more appropriate.

Reviewer 3 Report

Dear colleagues

It very exceptional, but I will accept the paper entitled “ FA Sliding as the Mechanism for the ANT1-Mediated Fatty Acid Anion Transport in Lipid Bilayers “ presented by  Jürgen Kreiter et al. without any modification. This article is ready for publication in IJMS as is come.

Author Response

We thank the reviewer for the positive comment.

Round 2

Reviewer 1 Report

The authors have addressed my questions. The manuscript can be published.